# Weather-responsive adaptive shading through biobased and bioinspired hygromorphic 4D-printing

Tiffany Cheng [1,2,8] ✉, Yasaman Tahouni [1,2,8], Ekin Sila Sahin [1,2], Kim Ulrich [3,4], Silvia Lajewski [5], Christian Bonten [5], Dylan Wood [1,2,6], Jürgen Rühe [3,7], Thomas Speck [3,4] & Achim Menges [1,2]

In response to the global challenge of reducing carbon emissions and energy consumption from regulating indoor climates, we investigate the applicability of biobased cellulosic materials and bioinspired 4D-printing for weather-responsive adaptive shading in building facades. Cellulose is an abundantly available natural material resource that exhibits hygromorphic actuation potential when used in 4D-printing to emulate motile plant structures in bioinspired bilayers. Three key aspects are addressed: (i) examining the motion response of 4D-printed hygromorphic bilayers to both temperature and relative humidity, (ii) verifying the responsiveness of self-shaping shading elements in lab-generated conditions as well as under daily and seasonal weather conditions for over a year, and (iii) deploying the adaptive shading system for testing in a real building facade by upscaling the 4D-printing manufacturing process. This study demonstrates that hygromorphic bilayers can be utilized for weather-responsive facades and that the presented system is architecturally scalable in quantity. Bioinspired 4D-printing and biobased cellulosic materials offer a resource-efficient and energy-autonomous solution for adaptive shading, with potential contributions towards indoor climate regulation and climate change mitigation.

The current climate crisis continues to pose a significant threat to our environment and quality of life, with the reduction of carbon dioxide ($CO_2$) emissions becoming an urgent global necessity in recent years[1]. Building operations to maintain indoor comfort are responsible for an estimated 27% of global $CO_2$ emissions[2], while manufacturing construction materials currently accounts for 11%[2]. Thus, the built environment accounts for 37% of the world's total emissions[2]. Building facades play a critical role in maintaining occupant comfort. Adaptive facades have the potential to reduce the need for active heating and cooling, thereby decreasing the energy consumption and environmental impact of buildings[3]. For example, adaptive shading systems can regulate the interior climate more efficiently by dynamically changing their orientation, shape, or thermal properties in response to external weather conditions.

However, typical approaches in kinetic shading systems often rely on operating energy to actuate electro-mechanical devices driving multiple moving parts. These complex assemblies are prone to malfunctions and require costly repairs to maintain[4]. Alternative systems

[1]Institute for Computational Design and Construction (ICD), University of Stuttgart, Stuttgart, Germany. [2]Cluster of Excellence IntCDC, University of Stuttgart, Stuttgart, Germany. [3]Cluster of Excellence livMatS @ FIT, University of Freiburg, Freiburg, Germany. [4]Plant Biomechanics Group, Botanic Garden, University of Freiburg, Freiburg, Germany. [5]Institut für Kunststofftechnik (IKT), University of Stuttgart, Stuttgart, Germany. [6]School of Architecture & Environment, University of Oregon, Eugene, USA. [7]Department for Microsystems Engineering (IMTEK), University of Freiburg, Freiburg, Germany. [8]These authors contributed equally: Tiffany Cheng, Yasaman Tahouni. ✉e-mail: tiffany.cheng@icd.uni-stuttgart.de

such as switchable ethylene-tetrafluoroethylene (ETFE) foils or passive tinted anti-sun glass have the disadvantage of filtering out too much daylight, often resulting in discomfort from poor lighting conditions or increased energy consumption for artificial indoor lighting[5]. Moreover, these systems are constructed with high embodied energy materials and manufacturing processes. The embodied carbon from manufacturing these construction materials can be reduced by shifting to biobased materials and material-efficient manufacturing methods[6].

Nature has provided solutions for adaptation, inspired by biological materials that respond passively to external stimuli. Motile plant organs, such as digging seeds[7–9] and pine cones[10–13], change their shape as completely dead material systems without any metabolic energy being involved. The structural basis of their autonomous adaptiveness lies in the arrangement of cellulose microfibrils, which cause anisotropic swelling of the cells and tissues due to moisture absorption from the air. The combination of highly swellable and less swellable tissue layers leads to an extraordinarily robust and resilient[14] bending reaction to changes in humidity[15,16]. This principle of self-shaping has been harnessed in bioinspired bilayer structures[17–22] composed of a swellable actuating layer and a relatively less swellable and perpendicularly aligned restricting layer.

Bilayers utilizing wood, a natural cellulose-based biocomposite, have been proposed for adaptive shading[23–26]. However, wood bilayers are difficult to engineer due to their intrinsic variability and manufacturing involving multiple manual steps. Advancements in 4D-printing[27] include the fused filament fabrication (FFF) of cellulosic thermoplastic bilayers[28], which has enabled more customizability in design[29] through the mesostructure[30]. For example, by controlling the extrusion process and resultant anisotropy and density of material paths[31], 4D-printed bilayers can be programmed to self-shape based on the functional principles of various biological role models[32–35]. While cellulosic materials have been further developed for humidity-responsiveness[36–40] and proposed for energy-autonomous shading[41–43], the architectural application of hygromorphic bilayers have so far been limited due to the lack of integrative research in their responsiveness to both temperature and humidity, robustness against weathering, and the feasibility of upscaling 4D-printing as a manufacturing process[44–46].

In this study, we investigate the applicability of biobased materials and bioinspired 4D-printing (Fig. 1a–c) for weather-responsive shading in building facades. As a case study, we design and develop the adaptive shading system in the context of the *liv*MatS Biomimetic Shell research building in Freiburg, Germany. We first examine the motion response of 4D-printed hygromorphic bilayers as a function of both temperature and relative humidity (RH), and we then verify the cyclic actuation and robustness of the 4D-printed bilayers against daily and seasonal weather effects (Fig. 1d–f). Finally, we deploy the 4D-printed shading system under real conditions at the scale of a building facade. By validating the weather-responsive shading system through a full-scale demonstrator (Fig. 1g–h), we highlight the potential of biobased cellulosic materials and bioinspired 4D-printing as energy-autonomous and resource-efficient solutions for decreasing the environmental impact of buildings.

## Results

### Hygromorphic materials for solar shading in a temperate climate

As a case study, the proposed weather-responsive shading system is demonstrated using the *liv*MatS Biomimetic Shell research building, located at the coordinates 48°00′53.6″N 7°50′03.1″E in Freiburg, Germany. This region of Southwest Germany is characterized by a temperate climate with dry and hot summers as well as humid and cold winters, creating an indirect but reliable link between the need for shading and the behavior of hygromorphic materials in response to RH. For example, summers here typically exhibit average daytime temperatures of 21 °C and 47% RH, while winters usually show average daytime temperatures of 5 °C and 82% RH (see "Methods": *Data*

*collection*). The motion response of adaptive shading elements should therefore close during low RH and high-temperature conditions to block out the sun, and open during high RH and low-temperature conditions, allowing the sun's heat to penetrate the building envelope. Additionally positioning the shading system south allows for greater harvesting of solar heat during the winter (Fig. 2a).

Considering these environmental and site conditions (see "Methods": *Digital design of adaptive shading elements*), we propose a modular design (Fig. 2d) based on a pair of self-shaping bilayer flaps that address the sun's east-to-west azimuth path, with changes in shape that adjust to the sun's elevation angles, which changes daily as well as seasonally from 18.5° at noon on the winter solstice (Fig. 2b) to 65° at noon on the summer solstice (Fig. 2c). The geometry of the flap elements can be strategically programmed to capture solar heat by curling to increase the opening area during cold days and months (Fig. 2e), while flattening to increase shade coverage and prevent excessive heat gain during warm days and months (Fig. 2f). These self-shaping shading elements are designed to be suspended within a double-skin facade with vents (see "Methods": *Facade system mockup*), allowing equalization to exterior weather conditions while being protected from direct exposure to water and rain.

### Influence of temperature on bilayer curvature

In order to understand the motion response of the self-shaping shading elements to a variety of weather conditions, we examine the curvature of 4D-printed cellulosic bilayers as a function of both RH and temperature. The cellulosic materials were produced (see "Methods": *Materials*, Supplementary Fig. 1), and the bilayer specimens prepared (see "Methods": *4D-Printing* and *Hygromorphic bilayers*). The bilayers' mesostructure design is illustrated in Fig. 3a. To mimic the orientation of cellulose microfibrils in pine cone scales, the actuating layer (layer 1) is formed by extruding the swellable biocomposite filament in the direction perpendicular to bending, while the restricting layer (layer 2) is formed by extruding the non-swellable ASA filament in the same direction of bending. The cross-patterned layer 3 is functionally inert and only exists to prevent delamination in our hygromorphic bilayers by sandwiching the restricting layer in between two layers made of the same biocomposite filament material, aiding in adhesion.

Under lab-generated test conditions (Fig. 4a), the bilayers were subjected to different permutations of RH and temperature (see "Methods": *Test series on temperature-dependence*) and their curvatures measured (Fig. 4b–c). While high RH is correlated to high curvatures, results show that the temperature also influences the trends at low RH. Figure 4d shows the mean and standard deviation across all bilayer specimens. At 80% RH, all bilayers are observed with an average curvature of 0.0155 mm$^{-1}$, regardless of the measured temperature. At a higher temperature of 30 °C, the cellulosic bilayers are observed to have almost no curvature at the lowest RH values. Conversely, at a lower temperature of 10 °C, the cellulosic bilayers exhibit relatively higher curvatures (0.011 mm$^{-1}$) even at the lower end of the RH spectrum. An intermediate temperature of 20 °C yields a curvature that falls between the two temperature values.

The influence of temperature on curvature is ideal for allowing increased light penetration on winter days, even under dry conditions. Similarly, during the occasional climatic occurrence of high humidity on a hot summer day due to summer thunderstorms, the cellulosic bilayers are biased toward a relatively closed configuration, aiding in shade coverage. The results of this study additionally reveal the extents of bilayer transformation (Fig. 3b), allowing us to better design the self-shaping shading element (Fig. 3c) and program its change in shape (Fig. 3d).

### Cyclic actuation, robustness against UV exposure, and actuation speed

Application in building systems necessitates that the adaptive shading elements perform reliably long-term, remain durable against the

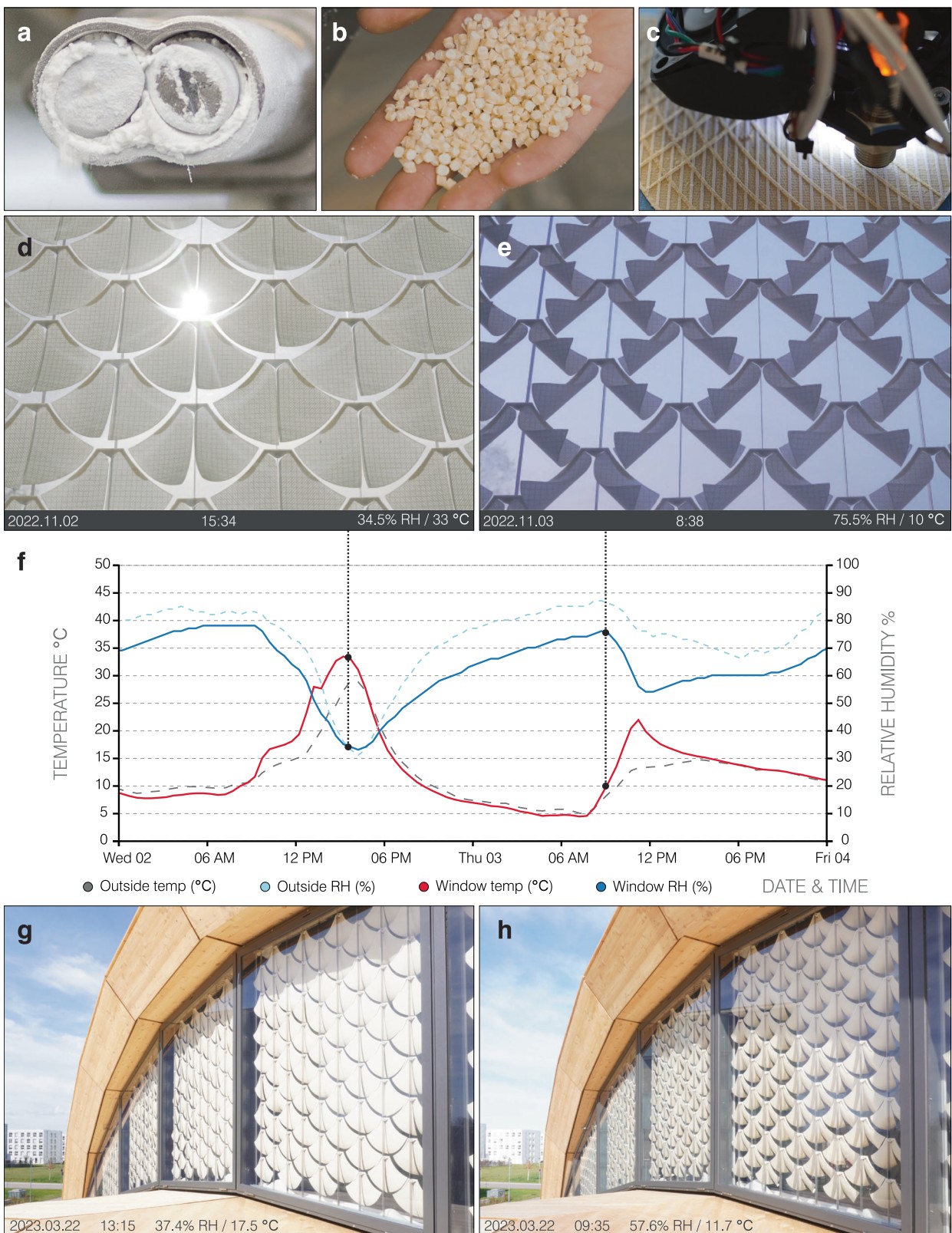

**Fig. 1 | An overview of the study on bioinspired and biobased weather-responsive shading. a** Cellulose is utilized for **b** hygromorphic filament materials. **c** Cellulosic bilayers, inspired by motile plant structures, are manufactured using 4D-printing. The 4D-printed shading elements are evaluated under real weather conditions, **d** flattening to increase shade coverage and **e** curling to increase opening area **f** on a particularly warm November 2nd and relatively cold November 3rd in 2022, respectively. The adaptive shading system is deployed on a building facade, **g** showing a closed configuration at 17.5 °C and **h** an open configuration at 11.7 °C.

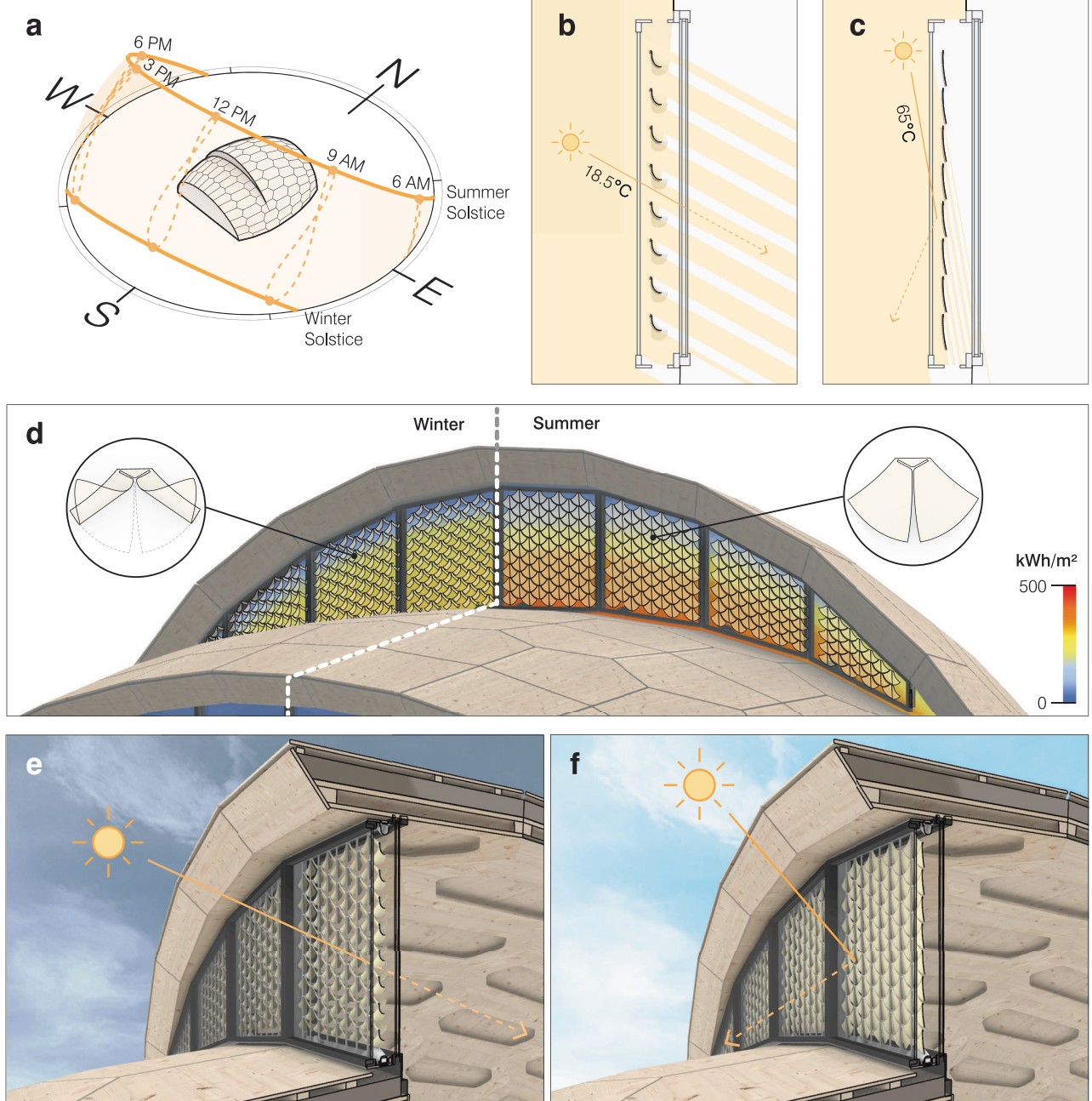

**Fig. 2 | The design of adaptive shading elements in the context of environmental and site conditions. a** The south-facing building facade **b** lets the low-angle sunlight pass through during winter, when the self-shaping bilayers are curled (shown at 18.5° during noon on the winter solstice). **c** During summer, the self-shaping bilayers flatten to block the high-angle sunlight (shown at 65° during noon on the summer solstice). **d** The modular design of the self-regulating shading element allows for **e** the maximum harvesting of solar heat during winter months and **f** shields high incident radiation from entering the building during summer months.

effects of UV exposure, and respond quickly to changes in RH for real-time adaptation to natural weather cycles.

We evaluated the longevity of 4D-printed cellulosic bilayers through multiple actuation cycles in a humidity-controlled environment (see "Methods": *Test series on cyclic actuation*). For all 170 tested actuation cycles, the bilayers display consistent curvatures at the low RH of 30%. At the high RH of 90%, there is some slight reduction in bilayer curvature within the first ten cycles of actuation; in the following 160 cycles, the bilayer curvatures remain generally consistent (Fig. 5c) regardless of the bilayers' orientation and support anchor (Fig. 5a).

To evaluate the robustness of the 4D-printed bilayers against UV exposure, we tested the specimens in a UV chamber (see "Methods": *Test series on UV exposure*). In general, UV-exposed bilayer specimens exhibit a similar motion response compared to the control bilayer specimens (Fig. 5d). At the high RH of 90%, the condition in which the bilayers are curled, UV exposure does not have a considerable effect on bilayer curvature. Bilayers exposed on both the active and restrictive layer sides (Fig. 5b) measured less than 0.001 mm$^{-1}$ difference in curvature compared to the control set. At the low RH of 30%, the condition in which the bilayers are flattened, the effect of UV exposure is more pronounced. Bilayers exposed on the restrictive layer side measured 0.002 mm$^{-1}$ reduced curvature, while bilayers exposed on the active layer side measured 0.006 mm$^{-1}$ reduced curvature. However, lower curvatures for shading are generally desired at lower RH conditions, when temperatures are also higher.

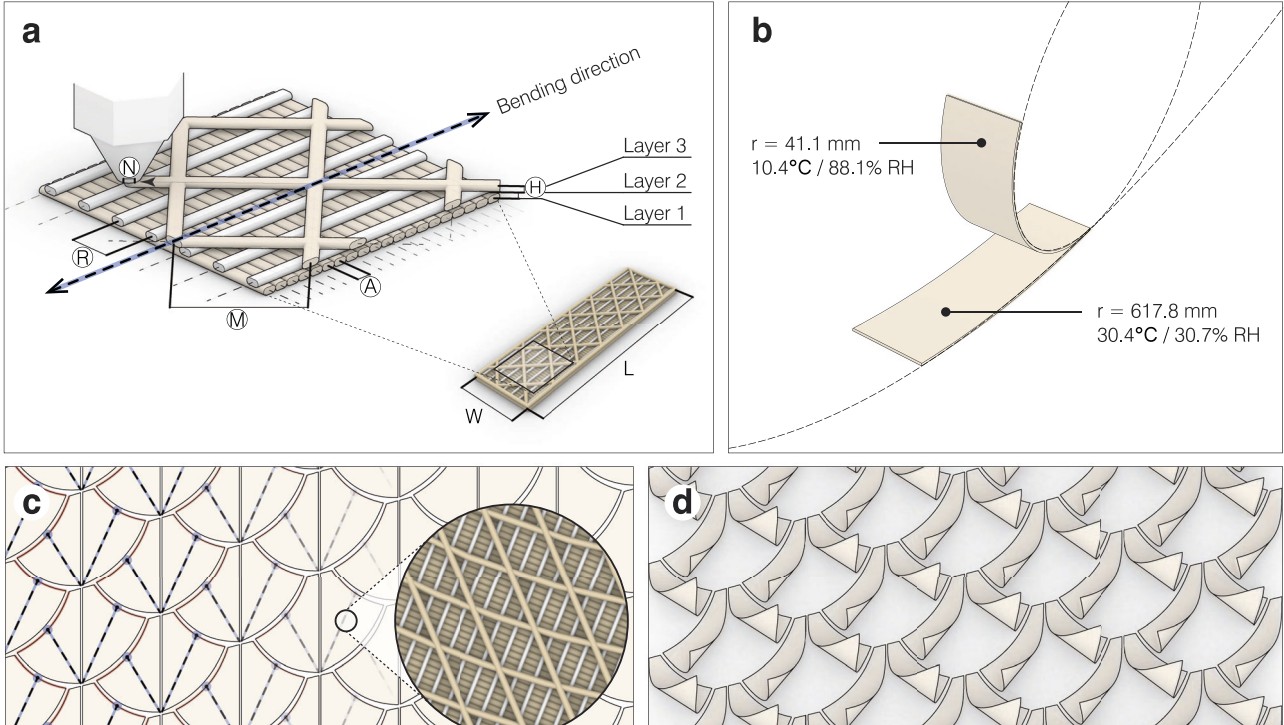

**Fig. 3 | The 4D-printing of self-shaping bilayers and adaptive shading elements.** **a** Illustrated here is the meso-scale structuring of the bilayers in this study, manufactured using a fused filament fabrication (FFF) 3D-printer with nozzle size $N$ and height of $H$ between the layers. The direction of bilayer bending (striped arrow) is programmed through the controlled paths of material deposition. The actuating layer (layer 1) is printed in the direction perpendicular to bending, using the biocomposite filament with an offset of $A$ between extruded paths, while the restricting layer (layer 2) is printed in the same direction as bending, using ASA filament with an offset of $R$ between extruded paths. Layer 3 is printed using the biocomposite filament in a criss-cross pattern with an offset of $M$ between extruded paths, aiding in bilayer adhesion by sandwiching the restricting layer between two layers made of the same material. **b** The resulting self-shaping bilayers are triggered by high RH to bend in the specified direction, with the highest curvatures measured at 88% RH and 10 ℃. **c** Each module in the designed tessellation is uniquely programmed to bend in the direction of the striped line, constructed from the flap tip to midpoint of the opposite edge. Through the 4D-printing process, the meso-scale structuring is applied to each module and **d** the adaptive shading elements curl in their designated directions.

To ensure suitable actuation speeds of the bilayers, we studied their responsiveness to desorption and absorption (see "Methods": *Test series on actuation speed*). The bilayers reach a full transformation between two RH extremes of 30% and 90% within 30 minutes (Fig. 5e). Considering the pace of RH fluctuations in natural weather cycles, our 4D-printed material system demonstrates sufficient responsiveness for the adaptive shading application.

### Responsiveness of self-shaping shading elements under real weather conditions

Following the tests under lab-generated conditions, we verify the 4D-printed shading elements under real weather conditions. A tessellated arrangement of the 4D-printed shading module was installed in a mock-up of the facade system and evaluated for 13 months under the full range of daily and seasonal weather effects (see "Methods": *Facade system mock-up* and *Data collection*, Supplementary Fig. 2 and *Analysis of shade coverage*, Supplementary Fig. 3).

The measured data from June 2022 to July 2023 shows a range of RH changes between 10% RH to 91% RH in daily and seasonal time scales, and the 4D-printed modules correspondingly open and close in response to these weather cycles. In the summer months, natural RH levels fluctuate significantly, and the 4D-printed mechanisms exhibit cyclic motion daily (Supplementary Movie 1–2). During hot and sunny days, the RH is typically low at around 25%, and the 4D-printed modules reach nearly 90% shade coverage (Fig. 6a). The RH level typically rises during the night and early morning hours, resulting in the 4D-printed modules opening with 50% shade coverage to let in the early morning daylight.

In the typically high levels of RH during the winter months, the 4D-printed modules remain open throughout the day (Supplementary Movie 3–4), maximizing the available solar heat. On the occasional sunny and warm winter day, a decrease in RH results in a slight closing of the 4D-printed modules from 50% to 65% shade coverage (Fig. 6b). However, considering the lowered angle of the sun in the winter months, the available solar rays still penetrate through the openings. The long-term monitoring of the shading elements under real conditions additionally showcases the durability and robustness of the 4D-printed cellulosic bilayers to natural weathering conditions, which do not undergo any noticeable reduction in actuation or exhibit any mechanical damage.

### Scalability of manufacturing for building facades

We demonstrate the adaptive shading system as a full-scale building facade in Freiburg, Germany (see "Methods": *Building facade demonstrator*, Supplementary Fig. 4).

The 4D-printing manufacturing process is time-, material-, and cost-efficient (Supplementary Table 3). A total of four FFF 3D-printers were employed to scale up the 4D-printing of the self-shaping modules through parallel production. Each module took between 25 and 32 minutes to 4D-print, depending on its size, and the four 3D-printers produced all of the 424 unique self-shaping modules in only 17 days. Using four 3D-printers (working 10 hours per day), the rate of 4D-printing manufacturing is 18.25 hm$^{-2}$. In total, 5.5 kg of the actuating filament and 0.7 kg of the restricting filament were consumed, which is 0.59 kg m$^{-2}$ and 0.07 kg m$^{-2}$ respectively. The pure cellulose powder used in the cellulosic filaments is also inexpensive at € 1.52 per kg.

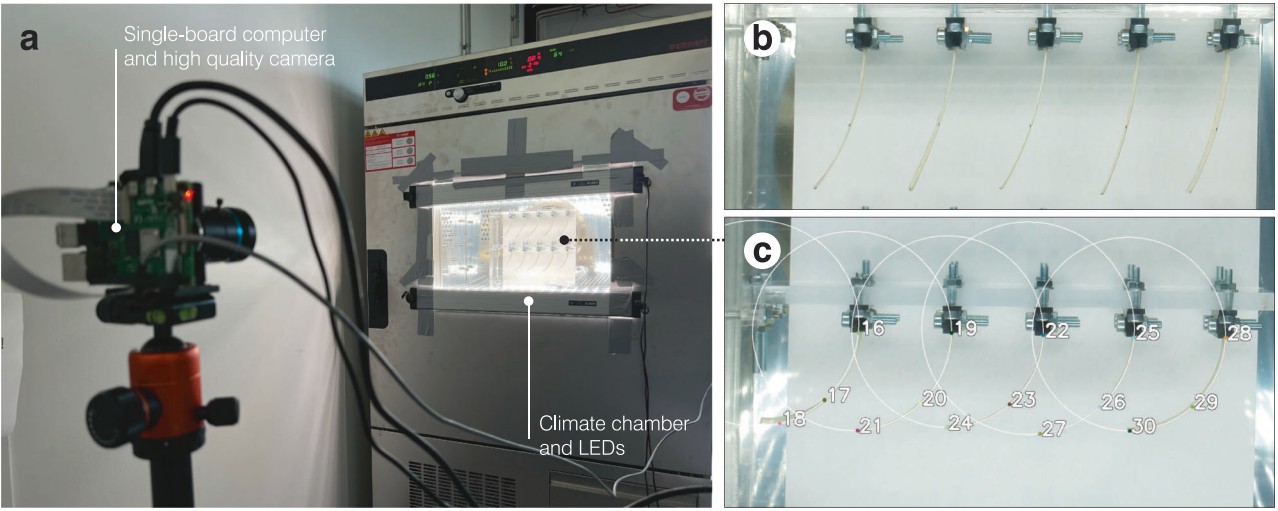

**Fig. 4 | Curvature of 4D-printed cellulosic bilayers as a function of relative humidity (RH) and temperature. a** Under lab-generated conditions of several RH and temperature combinations, **b** self-shaping bilayer specimens (*n* = 10) are recorded through the viewing window, **c** and their curvatures acquired using automated tracking. **d** For every image taken, the mean curvature was calculated across all specimens and plotted against the measured RH inside the climate chamber for each temperature tested (differentiated by color), with error bars representing the standard deviation of the mean curvature. At high RH, the bilayers measured consistent curvatures regardless of the temperature level. At low RH, results show almost no curvature at a higher temperature of 30 °C and relatively higher curvatures (0.011 mm⁻¹) at a lower temperature of 10 °C. An intermediate trend is observed at 20 °C. Source data are provided with this paper.

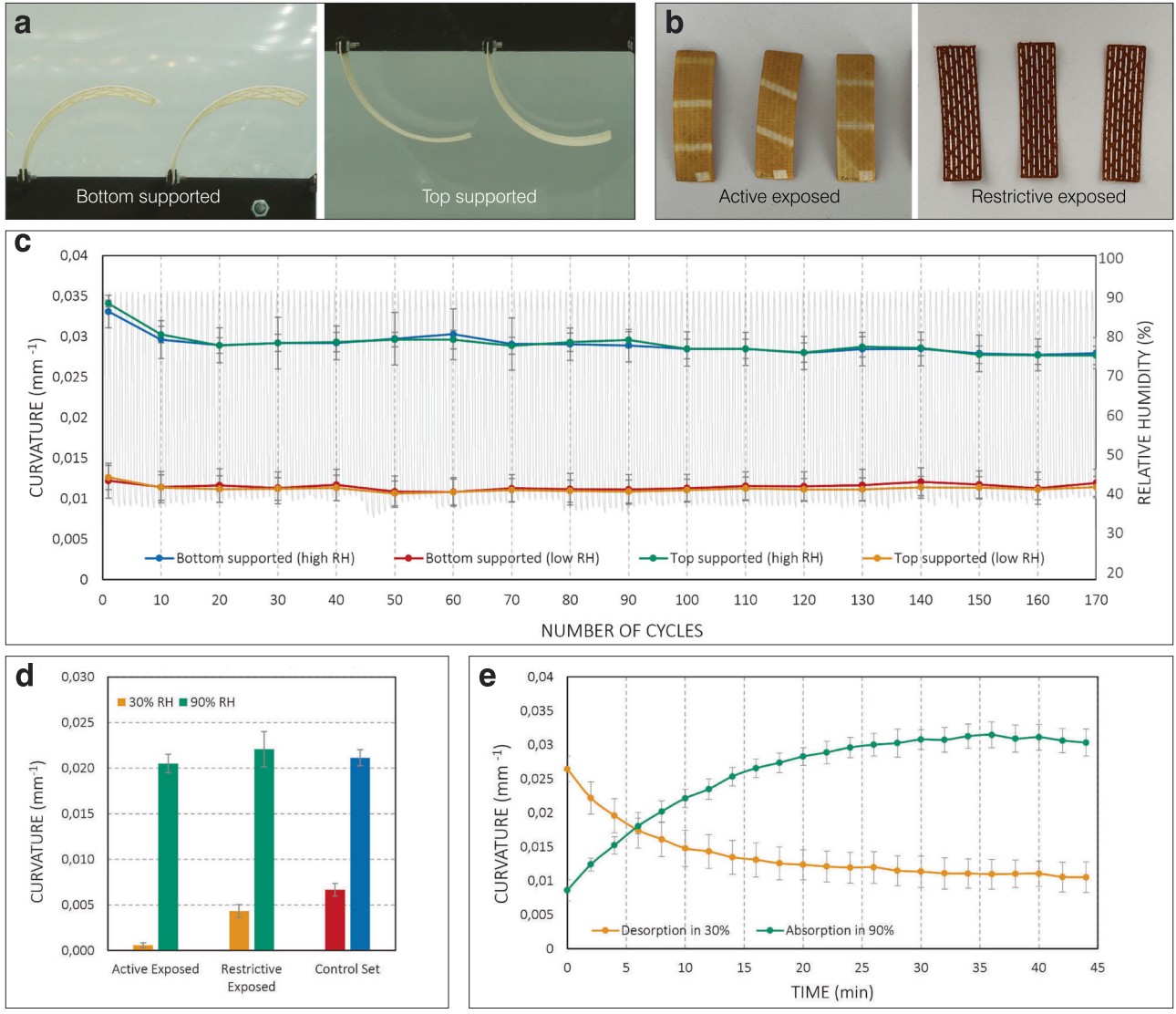

**Fig. 5 | Evaluation of cyclic actuation, UV exposure, and actuation speed.** The 4D-printed bilayers were tested **a** clamped at the bottom (left) and top (right), and **b** with the side of the active (left) and restrictive (right) layers up, shown here after UV exposure. **c** Curvature of the bilayer specimens ($n = 5$) at alternating RH levels from 30–90% for over 170 cycles of actuation in a humidity-controlled environment, demonstrating the reliability of the 4D-printed bilayers during cyclic actuation. **d** Impact of UV exposure on the curvature of the bilayer specimens ($n = 5$).

Results show that UV exposure marginally decreases curvature at low RH (30%), where a lower curvature is generally desired. UV exposure does not significantly affect the curvature at high RH (90%). **e** Curvature of the bilayer specimens ($n = 5$) over time in absorption (in 90% RH) and desorption (in 30% RH), indicating fast motion response of the 4D-printed bilayers. The error bars represent the standard deviation of the mean curvature.

Finally, the manufacturing process was fully streamlined with a minimal failure rate. Among 424 modules, only 30 modules needed to be re-printed due to minor flaws in their printing process (7% failure rate). After manufacturing, all modules were cycle tested for consistency of motion response in a humidity-controlled environment, after which only six modules were re-printed and replaced (1.4% failure rate). The final 4D-printed self-shaping modules were then deployed on the facade of the *liv*MatS Biomimetic Shell research building (Supplementary Fig. 5).

## Discussion

This study demonstrates the potential of biobased materials and bioinspired 4D-printing for weather-responsive adaptive shading in building facades. We determine hygromorphic materials to be suitable for solar shading in temperate climates by examining the influence of temperature as well as RH on the curvatures of 4D-printed cellulosic bilayers. Tested in lab-generated conditions, the motion response of

the 4D-printed bilayers is proven to remain robust against cyclic actuation and the effects of UV exposure. Future work will study in more detail the mechanism of UV-induced degradation and its long-term effects, as well as mechanical wear due to the actuation. The number of cycles tested is still small compared to the entire lifetime of a building, due to the physical timescale limits of humidity diffusion in the testing chamber as well as the poroelastic response time of the hygromorphic bilayers. However, in a mock-up of the facade system, the 4D-printed self-shaping elements have responded reliably to the daily and seasonal changes of real-world weather for over a year and will continue to be monitored long-term. While we have focused on evaluating the 4D-printed shading elements through analysis of shade coverage in this study, it will be important to quantify the improvement of indoor climate regulation in future work. As a case study and means of verification, we design and develop an adaptive shading system for the *liv*MatS Biomimetic Shell research building; the self-shaping shading elements, customized and programmed to match

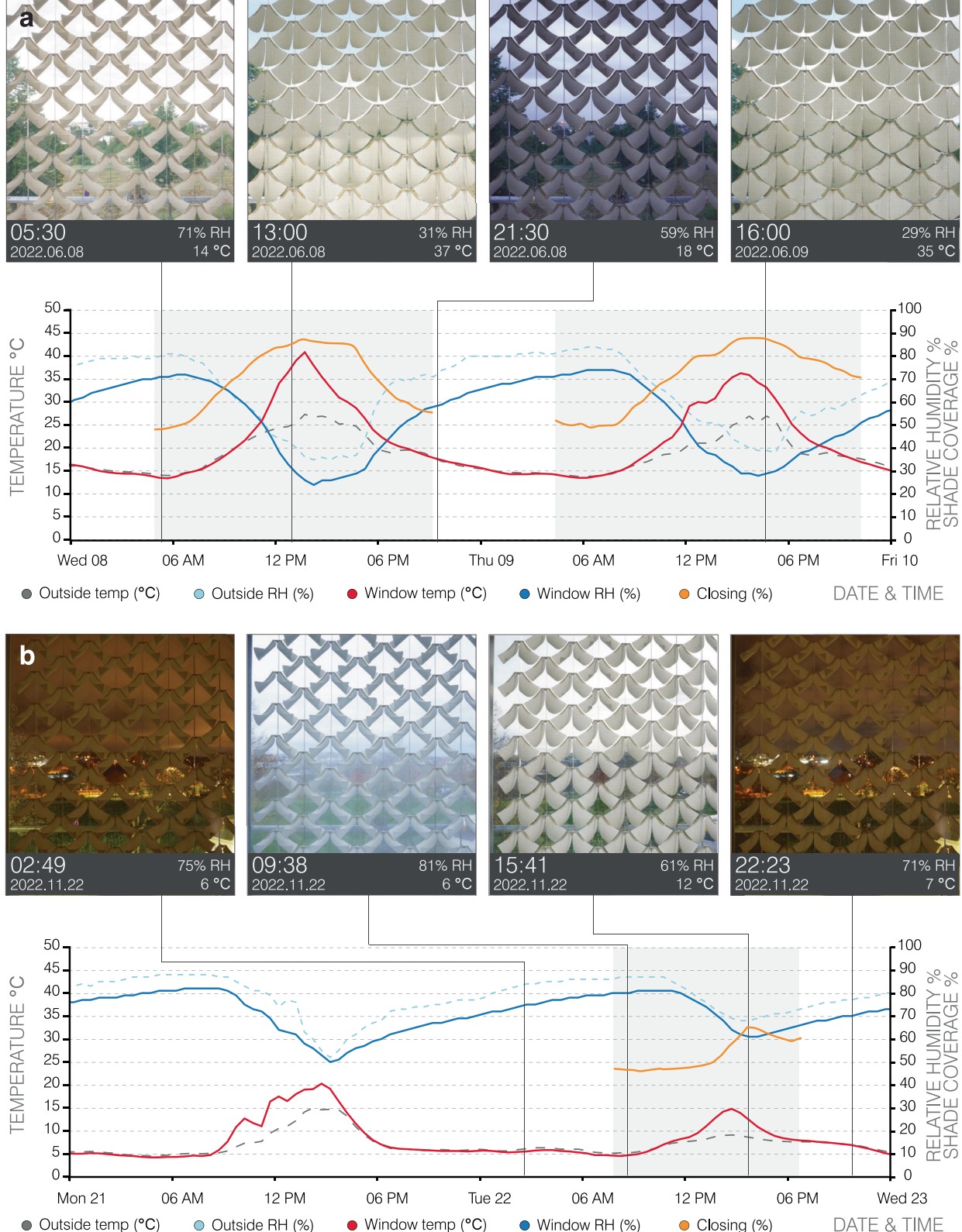

**Fig. 6 | The 4D-printed shading elements under real daily and seasonal weather conditions. a** During summer, significant daily oscillations in RH trigger closing with nearly 90% shade coverage during hot and sunny days with low RH, and opening with 50% shade coverage during nights and early morning hours with high RH. **b** During winter, the 4D-printed modules remain open throughout the typically high RH days, maximizing solar heat gain. On warmer winter days with relatively lower RH, the 4D-printed modules close slightly (from 50% to 65% shade coverage) while still allowing sun penetration due to the lowered sun angle. Source data are provided with this paper.

environment-, site-, and building-specific conditions, have functioned reliably for more than a year. Finally, upscaling the 4D-printing manufacturing process for a real building facade has demonstrated the feasibility, scalability, and practical implications of weather-responsive adaptive shading.

As the most abundant biomass on Earth, cellulose is a highly versatile and renewable natural material resource, making this work a potentially scalable solution for the global climate challenge of reducing energy consumption from regulating indoor climates. Designing with the inherent hygromorphic characteristics of cellulose, instead of highly refined smart materials, enables a sustainable and energy-autonomous solution to adaptive shading. Moreover, both the raw biobased materials and the FFF 3D-printers are highly accessible and affordable. Although our design approach has been based on an empirical process, the experiments have generated large amounts of image data, which can leverage machine learning to facilitate more accurate shape predictions and faster iteration in design with both forward and inverse design modeling approaches. The bioinspired 4D-printed shading system offers numerous advantages over current comparable electro-mechanical solutions in its low cost, low material consumption, and lightweight construction, as well as increased operational robustness due to the absence of moving mechanical parts. Furthermore, the strategic utilization of vents can be kept open to allow the adaptive shading elements to equalize with the environment or kept closed to override their passive response and provide agency to the building occupants through control of the self-shaping shading system.

The implications of this work extend beyond building facades, impacting broader areas of architecture, engineering, and construction. This research addresses the low productivity of the construction sector by infusing digital technology, new materials, and advanced automation into the design and engineering process[47]. The integration of environmental, site, building, material, and fabrication information in the digital model enables self-shaping elements to be tailored for specific functions. The feasibility of a bioinspired 4D-printing process has been demonstrated using readily available material resources (i.e., cellulose) and commonly available equipment (i.e., FFF 3D-printers). The functional architectural demonstrator not only proves the production to be scalable at cost but is also currently performing in a real-world context. It will be necessary to further improve the functionality of the weather-responsive adaptive shading by exploring applications in different building systems and climate zones, such as hot and humid tropical or cold and dry tundra climates. The 4D-printed shading system could be applied on larger glass surfaces, such as multi-story double-skin facade systems or stadium roofs[41], which have greater heating and cooling potentials. Future work will focus on investigating other types of suitable conditions in which 4D-printed hygromorphic systems could be utilized as interventions where cooling is urgently needed, such as in urban heat islands.

By embracing natural materials, bioinspired design principles, and leveraging digital manufacturing techniques, we can develop innovative solutions that address our environmental challenges while meeting the functional needs of our built environment. As we face increasing urgency to mitigate climate change impacts, the integration of such sustainable and adaptable solutions into our built environment will be critical in moving towards a more sustainable and resilient future.

## Methods

### Materials
The hygromorphic filament was produced from native cellulose powder (JELUCEL HM30; JELU-WERK; Rosenberg, Germany) as the filler (Supplementary Fig. 1a) and partially biobased polyketone (PK) (AKROTEK PK-VM natural; AKRO-PLASTIC; Niederzissen, Germany) as the matrix (Supplementary Fig. 1b). All raw materials were initially dried under vacuum to remove moisture, before compounding at a 35/65 mass ratio using a twin-screw extruder (ZSK26; Coperion; Stuttgart, Germany)

(Supplementary Fig. 1c). This mixture was then chopped into granules (Supplementary Fig. 1d) and dried under vacuum for 48 hours to remove any absorbed moisture. Subsequently, the dried granules were extruded into filaments with a consistent diameter of 1.75 mm[39,40] (LAB-Line; Collin; Maitenbeth, Germany) (Supplementary Fig. 1e). The filaments were spooled (Supplementary Fig. 1f) and sealed in vacuumed packaging foil, resulting in 16 kg of biocomposite filament for the actuating material layer. For the restricting material layer, an off-the-shelf acrylonitrile styrene acrylate (ASA) (ASA Natural; AzureFilm; Sežana, Slovenia) filament was chosen for its UV-resistant properties.

### 4D-printing
FFF 3D-printers with dual print heads (FELIX TEC 4; FELIXprinters; IJsselstein, Netherlands) were used to 4D-print the hygromorphic bilayers. The G-code machine instructions were generated using an integrated design-to-fabrication workflow[31] developed for a visual programming environment (Grasshopper 3D; version 6 SR13) running within a CAD 3D-modeling software (Rhinoceros 3D; Build 1.0.0007). Printing parameters were experimentally assigned, such as the temperature of the hot-ends as well as the feedrate, flowrate, and offset of extruded paths within each material layer. To ensure quality, consistency, and shape programmability of the 4D-printed bilayers, the printing environment was kept in dry conditions (30% RH, 20 ℃ to 23 ℃) and the hygroscopic filaments were printed from a dry box (Fiberthree Safe; Fiberthree; Darmstadt, Germany). Filaments were fed from directly above the machines to reduce the risk of breaking. The biocomposite filaments were extruded through a 0.7 mm hardened steel nozzle, and the ASA filaments were extruded through a 0.4 mm brass nozzle. After each print job, the biocomposite filament was immediately retracted, and the nozzle was purged using the ASA filament to prevent clogging. Painter's tape (Scotch-Blue; 3M; Minnesota, United States) was applied to the print bed for better adhesion. A separate FFF 3D-printer with a single print head (TAZ 6; LulzBot; Fargo, United States) was used to manufacture connectors for attaching the self-shaping bilayers to the facade system.

### Hygromorphic bilayers
Using a FFF 3D-printer to control the extrusion paths of actuating and restricting materials in two combined layers creates a moisture-driven bilayer system, similar to the structured tissue of a pine cone scale[12,13]. The mesostructure design, comprised of one actuating layer printed with the biocomposite filament and one restricting layer printed with the ASA filament, was optimized based on experimentation and extruded according to the specifications listed in Supplementary Table 1. Hygromorphic bilayer test specimens with the dimensions of 75 mm × 20 mm were 4D-printed eight at a time using one machine. The bilayer specimens were tested under lab-generated climate conditions in order to identify the extents of bilayer transformation at various temperature and RH combinations (see "Methods": *Test series on temperature-dependence*). With this information, we determined the maximum curvature in high RH conditions for designing the adaptive shading system (see "Methods": *Digital design of adaptive shading elements*), to be verified under real weather conditions (see "Methods": *Facade system mock-up*).

### Digital design of adaptive shading elements
The adaptive shading system was designed based on climate data for Freiburg, Germany (EnergyPlus Weather file retrieved from Climate.OneBuilding) using an environmental analysis plug-in (Ladybug; version 0.068) within the same 3D-modeling and visual programming environment as the 4D-printing workflow. Radiation studies on the case study building showed an average incident radiation of 270 *kWh/m²* during winter months, while the average incident radiation of 410 *kWh/m²* during the summer months. The geometry and motion response of the self-shaping shading elements were designed and iterated based on year-round performance evaluated through the difference in incident

radiation between the outside and inside glazing of the building's double-skin facade. A series of modular designs were analyzed with different objectives for the summer months (maximum difference when most flat to block solar heat) and winter months (minimum difference when most curled to harvest solar energy). The chosen double-flap module design was tessellated onto the building's facade, resulting in a collection of 424 unique modules. The specific geometries and motion responses of each double-flap module were then fed as inputs to the 4D-printing workflow, which automatically generated the 4D-printing paths of each material layer according to the curling direction of the bilayer flaps. The actuating layer material paths are oriented in the direction perpendicular to curling, while the restricting layer material paths are oriented in the same direction of curling.

## Test series on temperature-dependence

Ten bilayer specimens, arranged in two rows, were clamped to a custom-made specimen holder. The bilayers were clamped 10 mm at their upper ends, allowing 65 mm of the bilayers to move freely below. A black marker was used to mark three points along the visible edge of the bilayers for image tracking. The specimen holder, together with the ten bilayers, was positioned inside a climate chamber (CTC 256; Memmert GmbH + Co. KG; Schwabach, Germany) close to the viewing window. Two LED panels (HPB-60; AVOLUX SAS; Vendome, France) were additionally attached to the top and bottom of the viewing window for consistent lighting. In front of the viewing window, ~145 cm from the specimens, a small single-board computer (Raspberry Pi 4 Model B; Raspberry Pi Foundation; Cambridge, UK) with a camera module (Raspberry Pi High-Quality Camera; Raspberry Pi Foundation; Cambridge, UK) was set-up at the same height as the specimen holder (Fig. 4a). Connected to the Raspberry Pi was a temperature and humidity sensor (DHT22; Aosong Electronic Co., LTD.; Guangzhou, China) that was positioned next to the bilayers in the middle of the climate chamber and fixed to the specimen holder. The bilayer specimens were exposed to several permutations of temperature (10 °C, 20 °C, and 30 °C) and RH (in steps of 10% from 15–95% RH, according to allowable combinations of the climate chamber). For each test, the temperature was kept constant while each RH step was programmed for 2 hours when ascending (tested for all temperatures) and 3 hours when descending (tested for 30 °C). The Raspberry Pi simultaneously took photos and logged the temperature and humidity every 1-minute. Following image acquisition, we used a custom Python script (Python 3.11.4)[48] to select the three marked points along the visible edge of each bilayer, representing (1) the contact point between the bilayer specimen and the specimen holder, (2) the center of the visible edge, and (3) the tip of the bilayer edge. Once selected, the script then tracked the movement of these points across all images taken during the programmed temperature and humidity combinations, using the Lucas-Kanade Optical Flow algorithm of the OpenCV library (Version 4.7.0)[49]. After successful tracking, we calculated the radius and curvature of the circumcircle resulting from the three tracked points of each bilayer across all images. Since each image was accompanied by a temperature and humidity reading from our sensor inside the climate chamber, the arithmetic mean ( ± standard deviation) curvature across all ten bilayer samples was then calculated and plotted as a function of relative humidity. The temperature and humidity data of our sensor was used for visualization instead of the data from the climate chamber, since neither its programmed nor measured data represent the environment the specimens were exposed to precisely enough. Five sensor readings (out of the 2376 measured data points) showed to be faulty, since their temperature and humidity values exceeded what is possible for the climate chamber regulation. We therefore excluded this outlier data.

## Material characterization

To characterize the mechanical properties of materials used in this study, tensile bars were produced according to the ISO 527-2 Type 5A standard for molded and extruded plastics. Four sets of dumbbell-shaped test samples (seven samples per set) were 3D-printed. Each set was printed twice, once using the cellulosic filament and once with the ASA filament. Prior to performing the tensile tests, each set was conditioned as follows: Set no. 1 (control) kept at room conditions (40% RH, 20 °C to 23 °C); Set no. 2 (high RH) equalized at 90% RH for 24 hours; Set no. 3 (water-immersed) submerged in water for 12 hours; Set no. 4 (UV-exposed) exposed to a UV light source at a dry heat of 60 °C for 96 hours and kept at room conditions afterward. Tensile tests were performed using the 1476 universal testing machine from Zwick GmbH & Co. KG (Ulm, Germany) by applying a tensile force to the conditioned samples and measuring their properties until failure (Supplementary Table 2).

## Test series on cyclic actuation

Two sets of five bilayer specimens were clamped to a 3D-printed specimen holder arranged in two rows. To evaluate the effect of self-weight on long-term actuation and potential creep, Set no. 1 (bottom-supported) was clamped from the bottom and Set no. 2 (top-supported) was clamped from the top. The specimen holder and bilayers were placed inside a humidity-controlled chamber (MiniOne Humidity Generator; Preservatech; Bydgoszcz, Poland) programmed to cycle between 90% RH for 120 minutes and 30% RH for 240 minutes. The specimens were documented via time-lapse photography (Sony Alpha 6; Sony; Tokyo, Japan) at 1-minute intervals. The last image captured for each cycle's high RH and low RH was selected for curvature analysis using the ImageJ software[50]. For each image, three points (at the two ends and one in the middle) were manually clicked on the bilayer specimen to define a triangle, and the radius of the circumcircle was calculated using a script[51].

## Test series on UV exposure

Two sets of five bilayers were evaluated in a UV chamber under artificial weathering conditions according to the DIN EN ISO 4892-2/-3 UV standard for exposing plastics to large doses of irradiation with fluorescent UV and Xenon-arc lamps. The high intensity of light simulates UV weathering effects that occur in actual end-use environments at an accelerated timescale. The specimens were exposed to the UV light source at a dry heat of 60 °C for 96 hours from different orientations in relation to the UV light source. Set no. 1 (active-exposed) was exposed from the actuating layer side, and Set no. 2 (restrictive-exposed) was exposed from the restricting layer side. A control set (Set no. 3) was kept in room conditions, protected from UV exposure. All sets of bilayers were clamped to 3D-printed specimen holders and placed inside the humidity-controlled chamber. After 1 hour at 30% RH and at 90% RH, the bilayer specimens were photographed and their curvatures measured using ImageJ[50] and a script[51].

## Test series on actuation speed

Five bilayer specimens were documented for their responsiveness to desorption and absorption through time-lapse photography at 15-second intervals. For absorption, the specimens were initially equalized in a room environment (20 °C) with the RH level controlled at 30%, and then placed in the humidity-controlled chamber with 90% RH. For desorption, the specimens were equalized at 90% RH in the humidity-controlled chamber, and then moved to the room environment with 30% RH. Throughout the experiment, the RH and timestamp were logged at 1-minute intervals using the humidity-controlled chamber. Captured images were analyzed using ImageJ[50] and the bilayer curvatures calculated using a script[51].

## Facade system mock-up

We constructed a physical mock-up of the facade system in Stuttgart, Germany. The mock-up was positioned at a height (4 m) and orientation

(facing South, with an 11° offset to the West) that approximates the actual location and configuration of the building for which the shading system was designed, allowing for year-long simulation in comparable environmental conditions to Freiburg, Germany (Supplementary Fig. 2b). The mock-up comprised two modules, each with a 12 mm aluminum window box frame with an interior cavity measuring 1.5 cm × 12.1 cm × 14.9 cm for suspending an arrangement of the 4D-printed shading elements. Behind each window and separated by a double-glazed unit was an enclosed room dimensioned 13.5 cm × 11.8 cm × 19.5 cm with a door on the backside. Each window was enclosed with a framed, 4 mm polycarbonate (Markolon UV Clear 2099; Exolon Group; Pulheim, Germany) door on the front side and equipped with operable vents at the top and bottom of the box frame, which could be opened or sealed by an Arduino-compatible PLC (MINI; Controllino; Innsbruck, Austria).

### Data collection
The facade system mock-up was monitored inside the window boxes and rooms, as well as outside of the mock-up (Supplementary Fig. 2a), using weather-proof temperature and humidity sensors in mesh casing (SHT-30; Sensirion; Staefa, Switzerland). Two sets of the sensor were positioned inside each window box at the opposite corners. A sensor was mounted inside the ceiling of each room and outside of the mock-up under the roof (away from direct sunlight). An onboard computer (Raspberry Pi 3 Model B+; Raspberry Pi Foundation; Cambridge, UK) and generic USB webcam were used to remotely collect image and sensor data at 15-minute intervals as well as to open or seal the vents.

### Analysis of shade coverage
The 4D-printed shading elements inside the mock-up were captured using time-lapse photography (6000 × 4000 pixels, 350 dpi) for 28 weeks at 1-minute intervals. A custom Python script (Python 3.11.1) was used to filter the images and exclude photos that were too dark for data processing (from night to early morning). An image editor (Adobe Photoshop 2023) was used to crop filtered images to show a group of six complete 4D-printed modules in front of a clear sky background. These images were then converted into black and white, representing the 4D-printed module and sky background, respectively. The percentage of shade coverage was calculated using a custom Python script that analyzed the amount of black and white pixels (Supplementary Fig. 3).

### Building facade demonstrator
The design of the 4D-printed shading system was adapted from the facade system mock-up to the full-scale demonstrator in Freiburg, Germany (Supplementary Fig. 4). The target building facade consisted of eight geometrically unique windows covering a total area of 9.37 m². The aluminum window box frames, exchangeable plates with integrated vents for the top and bottom of each window, and mounting profiles for fastening the arrangement of shading elements were milled in-house with computer numerical control (CNC). The assembled window box frames were bolted to the structural steel frame of the double-glazed facade and enclosed with operable single-glazed glass doors with UV film (DK400; semaSORB; Coswig, Germany). Along with the 4D-printed shading elements, the same data collection infrastructure as the facade mock-up was installed in each window for long-term observation of the system.

## Data availability
Source data used in this study have been deposited in the Figshare database under accession code https://doi.org/10.6084/m9.figshare.26778376[52].

## Code availability
Computer code are provided at Code Ocean under accession code https://doi.org/10.24433/CO.6848806.v1[52].

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

## Acknowledgements

This research was funded by the Deutsche Forschungsgemeinschaft (DFG, German Research Foundation) under Germany's Excellence Strategy in the Clusters of Excellence IntCDC at the University of Stuttgart [EXC 2120/1-390831618; A.M., T.C., Y.T., D.W., E.S.S.] and livMatS @ FIT at the University of Freiburg [EXC 2193/1-390951807; J.R., T.S., K.U.]. Additional support was provided by the University of Stuttgart's "Internal Funding for Knowledge and Technology Transfer". The authors thank August Lehrecke, Fabian Eidner, Oliver Moldow, Selin Sevim, Aaron Wagner, Esra Yaman, Philipp Köser, Aleksa Arsic, Dennis Bartl, Sebastian Esser, Sven Gutekunst, Sven Hänzka, Sergej Klassen, Hendrik Köhler, Kai Stiefenhofer, Michael Preisack, and Michael Schneider who assisted in research tasks.

## Author contributions

Conceptualization: A.M., T.C., Y.T., E.S.S., D.W., T.S.; Methodology: T.C., Y.T., E.S.S., D.W., K.U., S.L.; Investigation: T.C., Y.T., E.S.S., K.U., S.L.; Formal Analysis: T.C., Y.T., E.S.S., K.U., S.L.; Writing (Original Draft): T.C., Y.T., E.S.S., K.U.; Writing (Review & Editing): T.C., Y.T., E.S.S., K.U., S.L., D.W., C.B., J.R., T.S., A.M.; Visualization: T.C., Y.T., E.S.S., K.U.; Supervision: A.M., T.S., J.R., D.W., C.B.; Resources: A.M., C.B., T.S., J.R.; Funding Acquisition: A.M., T.S., J.R., D.W., C.B.

## Funding

## Competing interests

The authors declare that they have no competing financial or non-financial interests.
