## [Transparent Peer Review file · Nature Communications]

Weather-Responsive Adaptive Shading through Biobased and Bioinspired Hygromorphic 4D-Printing

Corresponding Author: Ms Tiffany Cheng

Version 0:

Reviewer comments:

Reviewer #1

(Remarks to the Author)

The paper presents an interesting and novel idea of utilizing Bioinspired 4D-printing and biobased cellulosic materials as adaptive facades. Overall, this is a novel paper that highlights a new approach in the field.

The paper presented a series of experiments that took place to verify the results through the supplementary files. However, the paper is poorly structured and hard to follow, to begin with, the results followed by the methods. Also, the figures need to be more detailed to support the methods and the experiment process.

The method section needs to be elaborated more to give full details on the process. The study does not show exactly how the authors produced the filament, more details are required on this point as it is the core of the study. The paper did not also elaborate on the Python script and the OpenCV script used to track the material motion. The digital tracking part is not clear.

Reviewer #2

(Remarks to the Author)

In the manuscript "Weather-Responsive Adaptive Shading through Biobased and Bioinspired Hygromorphic 4D-Printing" Tiffany Cheng and colleagues present a partially bio-based shading system consisting of hygromorphic bilayers with a specific geometry that react upon changes in environmental conditions. The responsive elements are 3D-printed and incorporate anisotropic swelling and shrinkage through specifically printed structural patterns ("4D-printing"). This built-in responsiveness enables the modules to regulate shading autonomously, according to the environmental conditions; and within a reasonable time (~ 30 minutes).

The authors developed the 4D printed elements to a state where they could demonstrate a transfer to an application. The shading system was implemented in a new building in Freiburg, Germany. The climatic conditions of the city, the orientation of the building, as well as the solstices throughout the year, the eaves and the placement of the hygromorphic bilayers were all taken into account in the design of the building (for light and climate inside). This holistic approach is outstanding and has the potential to stimulate further discussion on sustainable architecture. Furthermore, the "experimental setting" in the form of a shading-system of a building allows to study the long-term ability of the structure for autonomous motion and it will be interesting to see if the buildings shading elements are still moving after 100s or 1000s of cycles.

Repeatability has been addressed in the paper. It is clear that 1000s of cycles in the lab are impossible, considering the response time of your elements. However, it is important to mention in relation to your experiments on the bilayers that the number of cycles performed is small compared to what is expected during the lifetime of a building.

Besides the effect of humidity on the curvature, effects of temperature and UV exposure were studied. We agree with the argument, that the effect of temperature is favorable for the application. We also understand that the benefit of a reduced curvature of the UV-treated sample at low relative humidities can be beneficial, but it is slightly contradictory for the low temperatures and the winter case. The paper would benefit from speculating on the possible effects of long-term UV exposure (degradation). Here we have to admit that we are not familiar with the standards (DIN, ISO, etc.) mentioned in the methods section. We assume that many readers will not have access to standards and therefore suggest that you briefly describe the experiments (Material characterisation, line 256, 262, UV exposure test series, line 312).

The data presented is substantial and valid. Interpretations and discussion are in line with the results and comprehensibly explained. Figures and tables are clear and informative. They contain the information needed to understand the investigation, starting from the behavior of the bi-layer over the "physical mock-up" long-term test, up to the final application of a real building facade. The supplementary videos nicely convey how the system works in application. The statistical

values shown in the graphs (data points and error bars) lack an explanation in the figure captions (median / mean, standard deviation / standard error / quartile?). The same applies for the mechanical data in supplementary table 2.

There is no information about the printing pattern used to create the partially cellulose-reinforced bilayer (extrusion direction and, if possible, cellulose orientation within the material). One can guess the structure of the rectangular bilayers (Suppl Fig. 1), but the print pattern of the shading elements remains completely unclear. Please add this information, as it is needed to understand how this work "addresses the key obstacles to digitalization in construction" (line 212), how the 3D geometries of the elements can change with environmental conditions, and to get an idea of the short response times. Explanations need to be added as well. Last but not least, this information is necessary for the reproducibility of the results.

Title, abstract and introduction are appropriate. The use of the term "bio-based" (title) is a bit optimistic as the largest proportion of the printed material is at best "partially bio-based" (line 229). Still, on our way towards a more sustainable world, it is reasonable to incorporate synthetic materials even to larger extents, as part of the transformation process.

Further comments:

Line 26: mentions indoor climate regulation by the shading system. However, no data is shown on how the indoor climate is improved. Please include some results based on the sensor data collected in the "physical mock-up"

Line 194f: How have the elements been customized and programmed to match all these conditions? What was the process? Which parameters were important? See also comment on missing information related to the design/structure of the shading elements.

Supplementary Table 2: Which kind of values are shown here? Are these representative measurements or is it statistical data? In the second case, which statistical value and what about the deviation?

Figures 3d, 4c-e: Which statistical values do the data points and error bars represent?

Reviewer #3

(Remarks to the Author)

Reviewer #4

(Remarks to the Author)

The authors employ 4D printed cellulosic materials to create adaptive shading in response to both temperature and relative humidity. Sufficient experimental data from temperature and humidity tests, cyclic tests, UV exposure tests, actuation speed tests are all provided to prove the efficiency and robustness in the application of weather-responsive shading. Overall, this is a novel and interesting work that potentially offers new possibility to 4D printing technology towards architecture area. For better quality of the manuscript, following comments should be further addressed:

1. The authors provide sufficient experimental data; however, the related mechanism is not clearly stated. What is the deformation mechanism of hygromorphic bilayer? What is the reason of giving a structural design shown in Supplementary Figure 1? Which layer is actuating layer, the perpendicular layer on the bottom or the cross-patterned layer on the top? Why the bottom cellulosic layer is perpendicular to the passive ASA filament layer?

2. The coupling between temperature and humidity is interesting and important for the bending of hygromorphic bilayer. In Figure 3d, under a given humidity (<75%), why does the bilayer structure demonstrate smaller curvature at higher temperature? And why does the effect of temperature diminish at greater humidity (>75%)?

3. In Figure 3d, there is no data for low temperature (10°C) and low humidity (40%). What is the reason? These data are also important because in northern part of China, the winter is cold and dry, which is different of the case in southern Germany where the winter is cold and humid. Please add comments.

4. Why does UV exposure have an obvious effect on the curvature at low humidity (30%), but negligible effect on the curvature at high humidity (90%)? Please explain the mechanism.

5. There are some literatures on the design of 4D printed structures (Nat Commun. 2024, 15 (1): 758; Forces in Mechanics. 2022, 7: 100081; Journal of Applied Mechanics. 2024, 91 (3): 030801). What is the design principle of the hygromorphic bilayer in the current manuscript? To realize a target deformed shape, how do the authors obtain the geometry parameters, a forward design method by developing analytical model or inverse design approach such as machine learning?

6. Some figures are not described in the main text including Figure 1a-1f and Figure 4a-4b.

The number of the figures should be labelled in an order. In the current manuscript, Supplementary Figure 4 appears prior to Supplementary Figure 1 in the main text.

Version 1:

Reviewer comments:

Reviewer #2

(Remarks to the Author)

Our concerns have been addressed appropriately - nice work!

Reviewer #3

(Remarks to the Author)

Reviewer #4

(Remarks to the Author)

The authors have well addressed my comments. I suggest the manuscript can be published as it is.

Response to the Reviewers

Reviewer #1 (R1)

The paper presents an interesting and novel idea of utilizing Bioinspired 4D-printing and biobased cellulosic materials as adaptive facades. Overall, this is a novel paper that highlights a new approach in the field.

R1.1: *The paper presented a series of experiments that took place to verify the results through the supplementary files. However, the paper is poorly structured and hard to follow, to begin with, the results followed by the methods. Also, the figures need to be more detailed to support the methods and the experiment process.*

Response: We structured our paper according to the house format of Nature Communications and will therefore keep the structure of the ‘Methods’ section following the ‘Results’ section. However, we have further developed the figures and their captions, in particular **Supplementary Fig. 1** (new figure) and **Fig. 3** (revised based on the former Supplementary Fig. 1), to show more details about the methods for filament production and digital design, respectively.

R1.2: *The method section needs to be elaborated more to give full details on the process. The study does not show exactly how the authors produced the filament, more details are required on this point as it is the core of the study. The paper did not also elaborate on the Python script and the OpenCV script used to track the material motion. The digital tracking part is not clear.*

Response: In order to cope with this suggestion, we have expanded upon the ‘Methods’ section. The filament production process was already described in detail in our previous papers (Kliem et al. 2020, *Fracture and Damage Mechanics*; Tahouni et al. 2022, *3D Printing and Additive Manufacturing*). To further illustrate this process, we have included a new figure (**Supplementary Fig. 1**) documenting each step and expanded the description about the filament production process (pages **11-12**, lines **256-266**).

- The hygromorphic filament was produced from native cellulose powder (JELUCEL HM30; JELU-WERK; Rosenberg, Germany) as the filler (Supplementary Fig. 1a) and partially biobased polyketone (PK) (AKROTEK PK-VM natural; AKRO-PLASTIC; Niedertzissen, Germany) as the matrix (Supplementary Fig. 1b). All raw materials were initially dried under vacuum to remove moisture, before compounding at a 35/65 mass ratio using a twin-screw extruder (ZSK26; Coperion; Stuttgart, Germany) (Supplementary Fig. 1c). This mixture was then chopped into granules (Supplementary Fig. 1d) and dried under vacuum for 48 hours to remove any absorbed moisture. Subsequently, the dried granules were extruded into filaments with a consistent diameter of 1.75mm (LAB-Line; Collin; Maitenbeth, Germany) (Supplementary Fig. 1e). The filaments were spooled (Supplementary Fig. 1f) and sealed in vacuumed packaging foil, resulting in 16 kg of biocomposite filament for the actuating material layer.

We have also added more information about the digital tracking procedure using Python and OpenCV (pages **14-15**, lines **333-343**).

- Following image acquisition, we used a custom Python script (Python 3.11.4) to select the three marked points along the visible edge of each bilayer, representing (1) the contact point between the bilayer specimen and the specimen holder, (2) the center of the visible edge, and (3) the tip of the bilayer edge. Once selected, the script then tracked the movement of these points across all images taken during the programmed temperature and humidity combinations, using the Lucas-Kanade Optical Flow algorithm of the OpenCV library (Version 4.7.0). After successful tracking, we calculated the radius and curvature of the circumcircle resulting from the three tracked points of each bilayer across all images. Since each image was accompanied by a temperature and humidity reading from our sensor inside the climate chamber, the arithmetic mean (\pm standard deviation) curvature across all ten bilayer samples was then calculated and plotted as a function of relative humidity.

Reviewer #2 (R2)

In the manuscript “Weather-Responsive Adaptive Shading through Biobased and Bioinspired Hygromorphic 4D-Printing” Tiffany Cheng and colleagues present a partially bio-based shading system consisting of hygromorphic bilayers with a specific geometry that react upon changes in environmental conditions. The responsive elements are 3D-printed and incorporate anisotropic swelling and shrinkage through specifically printed structural patterns (“4D-printing”). This built-in responsiveness enables the modules to regulate shading autonomously, according to the environmental conditions; and within a reasonable time (~ 30 minutes).

The authors developed the 4D printed elements to a state where they could demonstrate a transfer to an application. The shading system was implemented in a new building in Freiburg, Germany. The climatic conditions of the city, the orientation of the building, as well as the solstices throughout the year, the eaves and the placement of the hygromorphic bilayers were all taken into account in the design of the building (for light and climate inside). This holistic approach is outstanding and has the potential to stimulate further discussion on sustainable architecture. Furthermore, the “experimental setting” in the form of a shading-system of a building allows to study the long-term ability of the structure for autonomous motion and it will be interesting to see if the buildings shading elements are still moving after 100s or 1000s of cycles.

R2.1: *Repeatability has been addressed in the paper. It is clear that 1000s of cycles in the lab are impossible, considering the response time of your elements. However, it is important to mention in relation to your experiments on the bilayers that the number of cycles performed is small compared to what is expected during the lifetime of a building.*

Response: Indeed, the number of cycles tested is small compared to the entire lifetime of a building. We have expanded the ‘Discussion’ section to explain this limitation of the study (pages 9-10, lines 205-216).

- The number of cycles tested is small compared to the entire lifetime of a building, due to the physical timescale limits of humidity diffusion in the testing chamber as well as the poroelastic response time of the hygromorphic bilayers. However, in a mock-up of the facade system, the 4D-printed self-shaping elements have responded reliably to the daily and seasonal changes of real-world weather for over a year and will continue to be monitored long-term. . . . As a case study and means of verification, we design and

develop an adaptive shading system for the *livMatS* Biomimetic Shell research building; the self-shaping shading elements, customized and programmed to match environment-, site-, and building-specific conditions, have functioned reliably for more than a year.

R2.2: *Besides the effect of humidity on the curvature, effects of temperature and UV exposure were studied. We agree with the argument, that the effect of temperature is favorable for the application. We also understand that the benefit of a reduced curvature of the UV-treated sample at low relative humidities can be beneficial, but it is slightly contradictory for the low temperatures and the winter case. The paper would benefit from speculating on the possible effects of long-term UV exposure (degradation). Here we have to admit that we are not familiar with the standards (DIN, ISO, etc.) mentioned in the methods section. We assume that many readers will not have access to standards and therefore suggest that you briefly describe the experiments (Material characterisation, line 256, 262, UV exposure test series, line 312).*

Response: In order to cope with this suggestion, we have expanded the description of the experiments and standards for material characterization (page 15, lines 349-359) and UV exposure (page 16, lines 371-376).

- To characterize the mechanical properties of materials used in this study, tensile bars were produced according to the ISO 527-2 Type 5A standard for molded and extruded plastics. Four sets of dumbbell-shaped test samples (seven samples per set) were 3D-printed. Each set was printed twice, once using the cellulosic filament and once with the ASA filament. Prior to performing the tensile tests, each set was conditioned as follows: Set no. 1 (control) kept at room conditions (40% RH, 20 °C to 23 °C); Set no. 2 (high RH) equalized at 90% RH for 24 hours; Set no. 3 (water-immersed) submerged in water for 12 hours; Set no. 4 (UV-exposed) exposed to a UV light source at a dry heat of 60 °C for 96 hours and kept at room conditions afterward. Tensile tests were performed using the 1476 universal testing machine from Zwick GmbH & Co. KG (Ulm, Germany) by applying a tensile force to the conditioned samples and measuring their properties until failure (Supplementary Table 2).
- Two sets of five bilayers were evaluated in a UV chamber under artificial weathering conditions according to the DIN EN ISO 4892-2/-3 standard for exposing plastics to large doses of irradiation with fluorescent UV and Xenon-arc lamps. The high intensity of light simulates UV weathering effects that occur in actual end-use environments at an accelerated timescale. The specimens were exposed to the UV light source at a dry heat of 60 °C for 96 hours from different orientations in relation to the UV light source.

We studied the effects of UV exposure on the bilayers and found that the results were affected by which side of the bilayer (active or restrictive layer) was exposed (please see also response to point **R4.4** for our speculation on the effects of long-term UV exposure). The difference in curvature compared to the control set was minimal at 90% RH and notably reduced at 30% RH, a condition in which lower curvatures are desired for shading. We have elaborated the section ‘Cyclic actuation, robustness against UV exposure, and actuation speed’ to make this finding clearer (page 7, lines 143-152).

- In general, UV-exposed bilayer specimens exhibit a similar motion response compared to the control bilayer specimens (Fig. 5d). At the high RH of 90%, UV exposure does not

have a considerable effect on bilayer curvature. Bilayers exposed on both the active and restrictive layer sides measured less than $0,001 \text{ mm}^{-1}$ difference in curvature. At the low RH of 30%, the effect of UV exposure is more pronounced. Bilayers exposed on the restrictive layer side measured $0,002 \text{ mm}^{-1}$ reduced curvature, while bilayers exposed on the active layer side measured $0,006 \text{ mm}^{-1}$ reduced curvature. However, lower curvatures for shading are generally desired at lower RH conditions, when temperatures are also higher.

R2.3: *The data presented is substantial and valid. Interpretations and discussion are in line with the results and comprehensibly explained. Figures and tables are clear and informative. They contain the information needed to understand the investigation, starting from the behavior of the bi-layer over the “physical mock-up” long-term test, up to the final application of a real building facade. The supplementary videos nicely convey how the system works in application. The statistical values shown in the graphs (data points and error bars) lack an explanation in the figure captions (median / mean, standard deviation / standard error / quartile?). The same applies for the mechanical data in supplementary table 2.*

Response: To clarify how the graphs and data should be interpreted, we have added to the caption explanations in **Supplementary Table 2** (please see response to point **R2.8**), as well as **Fig. 4** (formerly Fig. 3) and **Fig. 5** (formerly Fig. 4) (please see response to point **R2.9**).

R2.4: *There is no information about the printing pattern used to create the partially cellulose-reinforced bilayer (extrusion direction and, if possible, cellulose orientation within the material). One can guess the structure of the rectangular bilayers (Suppl Fig. 1), but the print pattern of the shading elements remains completely unclear. Please add this information, as it is needed to understand how this work "addresses the key obstacles to digitalization in construction" (line 212), how the 3D geometries of the elements can change with environmental conditions, and to get an idea of the short response times. Explanations need to be added as well. Last but not least, this information is necessary for the reproducibility of the results.*

Response: We have expanded **Fig. 3** (formerly Supplementary Fig. 1) to show the connections between the 3D-printer's extruder, the rectangular bilayer, and the shading elements (please see also response to point **R4.1** for a more detailed explanation of the bilayer mechanism). To improve the reproducibility of the results, we have included more details about the digital design of the adaptive shading elements in the architectural and environmental contexts (please see also response to point **R2.7** for more details) and further expanded the 'Discussion' section on how this work addresses the obstacles to digitalization in construction (pages **10-11**, lines **235-242**).

- This research addresses the low productivity of the construction sector by infusing digital technology, new materials, and advanced automation into the design and engineering process. The integration of environmental, site, building, material, and fabrication information in the digital model enables self-shaping elements to be tailored for specific functions. The feasibility of a bioinspired 4D-printing process has been demonstrated using readily available material resources (i.e., cellulose) and commonly available equipment (i.e., FFF 3D-printers). The functional architectural demonstrator

not only proves the production to be scalable at cost but is also currently performing in a real-world context.

R2.5: *Title, abstract and introduction are appropriate. The use of the term “bio-based” (title) is a bit optimistic as the largest proportion of the printed material is at best “partially bio-based” (line 229). Still, on our way towards a more sustainable world, it is reasonable to incorporate synthetic materials even to larger extents, as part of the transformation process.*

Response: Thank you for this comment. Indeed, our material system is not fully biobased (yet). However, we completely agree that the pathway towards a more sustainable and decarbonized built environment involves a cumulative shift from synthetic to entirely biobased materials.

Further comments:

R2.6: *Line 26: mentions indoor climate regulation by the shading system. However, no data is shown on how the indoor climate is improved. Please include some results based on the sensor data collected in the “physical mock-up”*

Response: Our research investigated bioinspired 4D-printing and biobased materials for adaptive shading in building facades by evaluating the system’s response to daily and seasonal weather cycles. In this study, we have focused on monitoring the shape change of the adaptive shading elements through analysis of shade coverage. While quantifying the improvement of indoor climate regulation was not in the scope of this work, our physical mock-up has been designed and built for long-term monitoring of the indoor climate and comparisons to conventional shutter and shade systems. To clarify our work, we have adjusted this phrasing in the abstract (page 2, lines 25-27).

- Bioinspired 4D-printing and biobased cellulosic materials offer a resource-efficient and energy-autonomous solution for adaptive shading, with potential contributions towards indoor climate regulation and climate change mitigation.

R2.7: *Line 194f: How have the elements been customized and programmed to match all these conditions? What was the process? Which parameters were important? See also comment on missing information related to the design/structure of the shading elements.*

Response: The research building, *livMatS Biomimetic Shell*, was first analyzed based on its location and orientation on site. Several modular designs of the shading element were tessellated on the target façade and analyzed according to different objectives for the summer months and winter months. In summer, the tessellated design was analyzed in its flat state for low solar radiation passing through the façade. In winter, the tessellated design was analyzed in its curled state for high solar radiation passing through the façade. The selected module design was then programmed to curl in the specified direction, orientation, and radius through the material structuring created by the controlled extrusion of the 3D-printer. We further explain the designing and structuring of the shading elements in the methods (pages 13-14, lines 298-314).

- The adaptive shading system was designed based on climate data for Freiburg, Germany (EnergyPlus Weather file retrieved from Climate.OneBuilding) using an environmental

analysis plug-in (Ladybug; version 0.068) within the same 3D-modeling and visual programming environment as the 4D-printing workflow. . . . The geometry and motion response of the self-shaping shading elements were designed and iterated based on year-round performance evaluated through the difference in incident radiation between the outside and inside glazing of the building's double-skin facade. A series of modular designs were analyzed with different objectives for the summer months (maximum difference when most flat to block solar heat) and winter months (minimum difference when most curled to harvest solar energy). The chosen double-flap module design was tessellated onto the building's facade, resulting in a collection of 424 unique modules. The specific geometries and motion responses of each double-flap module were then fed as inputs to the 4D-printing workflow, which automatically generated the 4D-printing paths of each material layer according to the curling direction of the bilayer flaps. The actuating layer material paths are oriented in the direction perpendicular to curling, while the restricting layer material paths are oriented in the same direction of curling.

R2.8: *Supplementary Table 2: Which kind of values are shown here? Are these representative measurements or is it statistical data? In the second case, which statistical value and what about the deviation?*

Response: We have added the standard deviation from each experiment to **Supplementary Table 2** and revised the caption.

- **Supplementary Table 2:** Mechanical characterization of the biocomposite filament used for the actuating material layer. The values represent the mean and standard deviation of measurements from 7 samples per experiment.

R2.9: *Figures 3d, 4c-e: Which statistical values do the data points and error bars represent?*

Response: In **Fig. 4d** (formerly Fig. 3d), each point resembles the arithmetic mean across all ten bilayer specimens that were exposed to different temperature and humidity combinations, and the error bars are the corresponding standard deviation. To clarify this, we have revised the figure caption.

- **Figure 4:** Curvature of 4D-printed cellulosic bilayers as a function of relative humidity (RH) and temperature. (a) Under lab-generated conditions of several RH and temperature combinations, (b) self-shaping bilayer specimens ($n = 10$) are recorded through the viewing window, (c) and their curvatures acquired using automated tracking. (d) For every image taken, the mean curvature was calculated across all specimens and plotted against the measured RH inside the climate chamber for each temperature tested (differentiated by color), with error bars representing the standard deviation of the mean curvature. At high RH, the bilayers measured consistent curvatures regardless of temperature level. At low RH, results show almost no curvature at a higher temperature of 30 °C and relatively higher curvatures (0.011 mm^{-1}) at a lower temperature of 10 °C. An intermediate trend is observed at 20 °C.

Similarly, in **Fig. 5C-E** (formerly Fig. 4C-E), each point resembles the arithmetic mean across 5 samples tested per experiment, and the error bars are the corresponding standard deviation. This has been further clarified in the revised figure caption:

- **Figure 5:** Evaluation of cyclic actuation, UV exposure, and actuation speed. The 4D-printed bilayers were tested (a) clamped at the bottom (left) and top (right), and (b) with the side of the active (left) and restrictive (right) layers up, shown here after UV exposure. (c) Curvature of the bilayer specimens (n = 5) at alternating RH levels from 30-90% for over 170 cycles of actuation in a humidity-controlled environment, demonstrating the reliability of the 4D-printed bilayers during cyclic actuation. (d) Impact of UV exposure on the curvature of bilayer specimens (n = 5). Results show that UV exposure marginally decreases curvature at low RH (30%), where a lower curvature is generally desired. UV exposure does not significantly affect the curvature at high RH (90%). (e) Curvature of the bilayer specimens (n = 5) over time in absorption (in 90% RH) and desorption (in 30% RH), indicating the fast motion response of the 4D-printed bilayers.

Reviewer #3 (R3)

Reviewer #4 (R4)

The authors employ 4D printed cellulosic materials to create adaptive shading in response to both temperature and relative humidity. Sufficient experimental data from temperature and humidity tests, cyclic tests, UV exposure tests, actuation speed tests are all provided to prove the efficiency and robustness in the application of weather-responsive shading. Overall, this is a novel and interesting work that potentially offers new possibility to 4D printing technology towards architecture area. For better quality of the manuscript, following comments should be further addressed:

R4.1: *The authors provide sufficient experimental data; however, the related mechanism is not clearly stated. What is the deformation mechanism of hygromorphic bilayer? What is the reason of giving a structural design shown in Supplementary Figure 1? Which layer is actuating layer, the perpendicular layer on the bottom or the cross-patterned layer on the top? Why the bottom cellulosic layer is perpendicular to the passive ASA filament layer?*

Response: As explained in the introduction (page 3, lines 48-56), the mechanism of deformation is based on the material's swelling properties as well as the material's bilayer structure, inspired by natural hygromorphs such as pine cone scales. The opening and closing of pine cones are a prime example of hygroscopic movement in plants, which takes place in completely dead material systems without any metabolic energy being involved. The structural basis of their scales is two combined moisture-driven bilayer systems – each consisting of a lower, isotopically swellable tissue layer and an upper, longitudinally non-swellable tissue layer – which close under humid environmental conditions and open under dry conditions (*Ulrich et al. 2024, Bioinspiration & Biomimetics; Eger et al. 2022, Advanced Science*). This modular structure makes the opening mechanism of pine cones extraordinarily robust and resilient, as demonstrated by the fact that charcoalified pine cones are still functional after more than

100,000 years and closely related *Keteleeria* cones after more than 10 million years (Poppinga et al. 2017, *Scientific Reports*).

The extrusion paths of a fused filament fabrication 3D-printer can be controlled to tune the material deposition of the swellable and non-swellable filaments, creating a bilayer structure similar to the structured tissue of a pine cone scale. To clarify the mechanism of the hygromorphic bilayer in our manuscript, we have added further detail to **Fig. 3** (formerly Supplementary Fig. 1) and its caption, as well as the in-text explanation (page 5, lines **108-115**).

- The bilayers' mesostructure design is illustrated in Fig. 3a. To mimic the orientation of cellulose microfibrils in pine cone scales, the actuating layer (layer 1) is formed by extruding the swellable biocomposite filament in the direction perpendicular to bending, while the restricting layer (layer 2) is formed by extruding the non-swellable ASA filament in the same direction of bending. The cross-patterned layer 3 is functionally inert and only exists to prevent delamination in our hygromorphic bilayers by sandwiching the restricting layer in between two layers made of the same biocomposite filament material, aiding in adhesion.

R4.2: *The coupling between temperature and humidity is interesting and important for the bending of hygromorphic bilayer. In Figure 3d, under a given humidity (<75%), why does the bilayer structure demonstrate smaller curvature at higher temperature? And why does the effect of temperature diminish at greater humidity (>75%)?*

Response: The extent of swelling of the bilayer structures is controlled by two opposing parameters. There is on one side the Gibbs free energy of hydration, which allows to extract water molecules from the surrounding and bind them physically into the polymer structure. On the other side there is the energy of evaporation of the water back into the atmosphere. At a given temperature and humidity these two processes are in equilibrium. As the extent of swelling of the polymer is rather low entropy changes occurring when the swelling is changed are low and accordingly entropy-terms can be neglected and the free energy of hydration can be replaced by the enthalpy of hydration.

When the hygromorphic bilayers are placed inside a closed window cavity, the absolute humidity inside remains constant. As the temperature is raised, evaporation takes over and water from the actuating layer evaporates into the gas phase. This layer dries and accordingly shrinks. When the temperature cools back down, the enthalpy of hydration wins over the enthalpy of evaporation, water is bound by the polymer molecules and thus removed from the gas phase.

The set-point of the switching process can be adjusted by manipulating the humidity inside of the window cavity by temporarily opening and closing the vents, a topic we wish to investigate further in future work, as mentioned in the discussion (page 10, lines **230-233**). When the window is not completely hermetically closed, some small exchange of the air with the outside air occurs. It will then slowly move to a drifting average.

We give in our paper always relative humidities. This is easy to measure and describes the data very well as long as any temperature changes are small. From a principle point of view, the absolute humidity should be used as this is independent of the temperature.

R4.3: In Figure 3d, there is no data for low temperature (10°C) and low humidity (40%). What is the reason? These data are also important because in northern part of China, the winter is cold and dry, which is different of the case in southern Germany where the winter is cold and humid. Please add comments.

Response: Here we studied all conditions that can be obtained in our climate chamber, and the conditions to which our demonstrator building is exposed to are well within the climate chamber's working range. At 10 °C, the lowest RH that our climate chamber can reach is 40% RH. As mentioned in the discussion (page 11, lines 242-245), we are also interested in exploring additional climate zones in our future work, such as the hot and humid tropics or cold and dry tundra.

- It will be necessary to further improve the functionality of the weather-responsive adaptive shading by exploring applications in different building systems and climate zones, such as hot and humid tropical or cold and dry tundra climates.

R4.4: Why does UV exposure have an obvious effect on the curvature at low humidity (30%), but negligible effect on the curvature at high humidity (90%)? Please explain the mechanism.

Response: Exposure to high levels of UV radiation could potentially lead to polymer chain breakage and an increase of crosslinking, which would reduce the difference in bending. The experiments show that, after conditioning in the accelerated UV weathering test, no changes in bending curvature occur at a high relative humidity, independent of from which side irradiation was performed. At low levels of relative humidity, a small decrease of the curvature is overserved. We could speculate that at low relative humidity and therefore low degrees of swelling, the polymer chains are in very close contact, so that the generation of any chain breakages will lead to a slight increase in crosslinking. At higher relative humidity, however, the reactive species generated after UV-exposure could also react with water present in the film, leading to a smaller extent of crosslinking and thus lower changes. As indicated now in the discussion (page 9, lines 204-205), we are currently studying the mechanism of UV-induced degradation in more detail to support this hypothesis. Here it was just the goal of the study to demonstrate that the system can survive even quite strong UV-irradiation (much stronger than ever observed in a real building) in such an accelerated test, which is indeed shown in the experiments.

- Future work will study in more detail the mechanism of UV-induced degradation and its long-term effects, as well as mechanical wear due to the actuation.

R4.5: There are some literatures on the design of 4D printed structures (*Nat Commun.* 2024, 15 (1): 758; *Forces in Mechanics.* 2022, 7: 100081; *Journal of Applied Mechanics.* 2024, 91 (3): 030801). What is the design principle of the hygromorphic bilayer in the current manuscript? To realize a target deformed shape, how do the authors obtain the geometry parameters, a forward design method by developing analytical model or inverse design approach such as machine learning?

Response: The design principle of our hygromorphic bilayers (detailed in point R4.1) is based on many iterations of experimentation to identify the 4D-printing parameters for achieving desired shape changes. Similar to research reviewed in *Yuan et al.* 2022, *Forces in Mechanics*,

we conducted tests for the development of a forward design approach. We evaluated our 4D-printed bilayers in a climate chamber, studying their response to both temperature and humidity changes, and obtained their maximum and minimum curvatures at high and low RH, respectively. This empirical data was used for designing the adaptive shading module according to its radiation performance. The selected module design was manufactured with 4D-printing and installed in the façade mock-up for further verification under real weather conditions. This has been further clarified in the methods (page 13, lines 288-297).

- The mesostructure design, comprised of one actuating layer printed with the biocomposite filament and one restricting layer printed with the ASA filament, was optimized based on experimentation and extruded according to the specifications listed in Supplementary Table 1. . . . The bilayer specimens were tested under lab-generated climate conditions in order to identify the extents of bilayer transformation at various temperature and RH combinations (see Methods: Test series on temperature-dependence). With this information, we determined the maximum curvature in high RH conditions for designing the adaptive shading system (see Methods: Digital design of adaptive shading elements), to be verified under real weather conditions (see Methods: Facade system mock-up).

As now indicated in the discussion (page 10, lines 224-227), we see potential in employing machine learning to analyze our data from this research and correlate environmental factors with bilayer curvatures. Future work will focus on predicting curvature more accurately from the mesostructure design through a forward design approach. We are also interested in inverse design approaches, similar to research reviewed in *Sun et al. 2024, Journal of Applied Mechanics*, for determining the mesostructure design from desired shading geometries or performance requirements. Additionally, integrating pixel-based shade cover analysis (page 18, lines 411-419) and voxel-based 4D-printing design methods (*Sahin et al. 2023, Biomimetics*) will help predict shape change and improve real-world performance.

- Although our design approach has been based on an empirical process, the experiments have generated large amounts of image data, which can leverage machine learning to facilitate more accurate shape predictions and faster iteration in design with both forward and inverse design modeling approaches.

R4.6: *Some figures are not described in the main text including Figure 1a-1f and Figure 4a-4b. The number of the figures should be labelled in an order. In the current manuscript, Supplementary Figure 4 appears prior to Supplementary Figure 1 in the main text.*

Response: Thank you for pointing this out. We have now made sure that all main and supplementary figures are referenced in the text and are listed in the correct order.